# Convergence Analysis of Prediction Markets via Randomized Subspace Descent

**Rafael Frongillo**
Department of Computer Science
University of Colorado, Boulder
`raf@colorado.edu`

**Mark D. Reid**
Research School of Computer Science
The Australian National University & NICTA
`mark.reid@anu.edu.au`

## Abstract

Prediction markets are economic mechanisms for aggregating information about future events through sequential interactions with traders. The pricing mechanisms in these markets are known to be related to optimization algorithms in machine learning and through these connections we have some understanding of how equilibrium market prices relate to the beliefs of the traders in a market. However, little is known about rates and guarantees for the convergence of these sequential mechanisms, and two recent papers cite this as an important open question.

In this paper we show how some previously studied prediction market trading models can be understood as a natural generalization of randomized coordinate descent which we call *randomized subspace descent* (RSD). We establish convergence rates for RSD and leverage them to prove rates for the two prediction market models above, answering the open questions. Our results extend beyond standard centralized markets to arbitrary trade networks.

## 1    Introduction

In recent years, there has been an increasing appreciation of the shared mathematical foundations between prediction markets and a variety of techniques in machine learning. Prediction markets consist of agents who trade securities that pay out depending on the outcome of some uncertain, future event. As trading takes place, the prices of these securities reflect an aggregation of the beliefs the traders have about the future event. A popular class of mechanisms for updating these prices as trading occurs has been shown to be closely related to techniques from online learning [7, 1, 21], convex optimization [10, 19, 13], probabilistic aggregation [24, 14], and crowdsourcing [3]. Building these connections serve several purposes, however one important line of research has been to use insights from machine learning to better understand how to interpret prices in a prediction market as aggregations of trader beliefs, and moreover, how the market together with the traders can be viewed as something akin to a distributed machine learning algorithm [24].

The analysis in this paper was motivated in part by two pieces of work that considered the equilibria of prediction markets with specific models of trader behavior: traders as risk minimizers [13]; and traders who maximize expected exponential utility using beliefs from exponential families [2]. In both cases, the focus was on understanding the properties of the market at convergence, and questions concerning whether and how convergence happened were left as future work. In [2], the authors note that "we have not considered the dynamics by which such an equilibrium would be reached, nor the rate of convergence etc., yet we think such questions provide fruitful directions for future research." In [13], "One area of future work would be conducting a detailed analysis of this framework using the tools of convex optimisation. A particularly interesting topic is to find the conditions under which the market will converge."

The main contribution of this paper is to answer these questions of convergence. We do so by first proposing a new and very general model of trading networks and dynamics (§3) that subsumes the models used in [2] and [13] and provide a key structural result for what we call *efficient* trades in these networks (Theorem 2). As an aside, this structural result provides an immediate generalization of an existing aggregation result in [2] to trade networks of "compatible" agents (Theorem 8). In §4, we argue that efficient trades in our networks model can be viewed as steps of what we call *Random Subspace Descent* (RSD) algorithm (Algorithm 1). This novel generalization of coordinate descent allows an objective to be minimized by taking steps along affinely constrained subspaces, and maybe be of independent interest beyond prediction market analysis. We provide a convergence analysis of RSD under two sets of regularity constraints (Theorems 3 & 9) and show how these can be used to derive (slow & fast) convergence rates in trade networks (Theorems 4 & 5).

Before introducing our general trading networks and convergence rate results, we first introduce the now standard presentation of *potential-based prediction markets* [1] and the recent variant in which all agents determine their trades using *risk measures* [13]. We will then state informal versions of our main results so as to highlight how we address issues of convergence in existing frameworks.

## 2 Background and Informal Results

Prediction markets are mechanisms for eliciting and aggregating distributed information or beliefs about uncertain future events. The set of events or *outcomes* under consideration in the market will be denoted $\Omega$ and may be finite or infinite. For example, each outcome $\omega \in \Omega$ might represent a certain presidential candidate winning an election, the location of a missing submarine, or an unknown label for an item in a data set. Following [1], the goods that are traded in a prediction market are $k$ outcome-dependent *securities* $\{\phi(\cdot)_i\}_{i=1}^k$ that pay $\phi(\omega)_i$ dollars should the outcome $\omega \in \Omega$ occur. We denote the set of distributions over $\Omega$ by $\Delta_\Omega$ and note, for any $p \in \Delta_\Omega$, that the expected pay off for the securities under $p$ is $\mathbb{E}_{\omega \sim p}[\phi(\omega)]$ and the set of all expected pay offs is just the convex hull, denoted $\Pi := \text{conv}(\phi(\Omega))$. A simple and commonly studied case is when $\Omega = [k] := \{1, \ldots, k\}$ (*i.e.*, when there are exactly $k$ outcomes) and the securities are the *Arrow-Debreu* securities that pay out \$1 should a specific outcome occur and nothing otherwise (*i.e.*, $\phi(\omega)_i = 1$ if $\omega = i$ and $\phi(\omega)_i = 0$ for $\omega \neq i$). Here, the securities are just basis vectors for $\mathbb{R}^k$ and $\Pi = \Delta_\Omega$.

Traders in a prediction market hold portfolios of securities $r \in \mathbb{R}^k$ called *positions* that pay out a total of $r \cdot \phi(\omega) = \sum_{i=1}^k r_i \phi(\omega)_i$ dollars should outcome $\omega$ occur. We denote the set of positions by $\mathcal{R} = \mathbb{R}^k$. We will assume that $\mathcal{R}$ always contains a position $r^\$$ that returns a dollar regardless of which outcome occurs, meaning $r^\$ \cdot \phi(\omega) = 1$ for all $\omega \in \Omega$. We therefore interpret $r^\$$ as "cash" within the market in the sense that buying or selling $r^\$$ guarantees a fixed change in wealth.

In order to address the questions about convergence in [2, 13] we will consider a common form of prediction market that is run through a *market maker*. This is an automated agent that is willing to buy or sell securities in return for cash. The specific and well-studied prediction market mechanism we consider is the *potential-based market maker* [1]. Here, traders interact with the market maker sequentially, and the cost for each trade is determined by a convex potential function $C : \mathcal{R} \to \mathbb{R}$ applied to the market maker's *state* $s \in \mathcal{R}$. Specifically, the cost for a trade $dr$ when the market maker has state $s$ is given by $\text{cost}(dr; s) = C(s - dr) - C(s)$, *i.e.*, the change in potential value of the market maker's position due to the market maker accepting the trade. After a trade, the market maker updates the state to $s \leftarrow s - dr$.[1] As noted in the next section, the usual axiomatic requirements for a cost function (*e.g.*, in [1]) specify a function that is effectively a *risk measure*, commonly studied in mathematical finance (see, *e.g.*, [9]).

### 2.1 Risk Measures

As in [13], agents in our framework will each quantify their uncertainty in positions using what is known as risk measure. This is a function that assigns dollar values to positions. As Example 1 below shows, this assumption will also cover the case of agents maximizing exponential utility, as considered in [2].

A (convex monetary) *risk measure* is a function $\rho : \mathcal{R} \to \mathbb{R}$ satisfying, for all $r, r' \in \mathcal{R}$:

- *Monotonicity*: $\quad \forall \omega \quad r \cdot \phi(\omega) \leq r' \cdot \phi(\omega) \implies \rho(r) \geq \rho(r')$.
- *Cash invariance*: $\quad \rho(r + c\,r^{\$}) = \rho(r) - c \quad$ for all $c \in \mathbb{R}$.
- *Convexity*: $\quad \rho\big(\lambda r + (1 - \lambda)r'\big) \leq \lambda \rho(r) + (1 - \lambda)\rho(r') \quad$ for all $\lambda \in (0, 1)$.
- *Normalization*: $\quad \rho(0) = 0$.

The reasonableness of these properties is usually argued as follows (see, *e.g.*, [9]). Monotonicity ensures that positions that result in strictly smaller payoffs regardless of the outcome are considered more risky. Cash invariance captures the idea that if a guaranteed payment of $\$c$ is added to the payment on each outcome then the risk will decrease by $\$c$. Convexity states that merging positions results in lower risk. Finally, normalization requires that holding no securities should carry no risk. This last condition is only for convenience since any risk without this condition can trivially have its argument translated so it holds without affecting the other three properties. A key result concerning convex risk measures is the following representation theorem (cf. [9, Theorem 4.15], ).

**Theorem 1** (Risk Representation). *A functional $\rho : \mathcal{R} \to \mathbb{R}$ is a convex risk measure if and only if there is a closed convex function $\alpha : \Pi \to \mathbb{R} \cup \{\infty\}$ such that $\rho(r) = \sup_{\pi \in \mathrm{relint}(\Pi)} \langle \pi, -r \rangle - \alpha(\pi)$.*

Here $\mathrm{relint}(\Pi)$ denotes the relative interior of $\Pi$, the interior relative to the affine hull of $\Pi$. Notice that if $f^*$ denotes the convex conjugate $f^*(y) := \sup_x \langle y, x \rangle - f(x)$, then this theorem states that $\rho(r) = \alpha^*(-r)$, that is, $\rho$ and $\alpha$ are "dual" in the same way prices and positions are dual [5, §5.4.4]. This suggests that the function $\alpha$ can be interpreted as a *penalty function*, assigning a measure of "unlikeliness" $\alpha(\pi)$ to each expected value $\pi$ of the securities defined above. Equivalently, $\alpha(\mathbb{E}_p[\phi])$ measures the unlikeliness of distribution $p$ over the outcomes. We can then see that the risk is the greatest expected loss under each distribution, taking into account the penalties assigned by $\alpha$.

**Example 1.** *A well-studied risk measure is the* entropic risk *relative to a reference distribution $q \in \Delta_\Omega$ [9]. This is defined on positions $r \in \mathcal{R}$ by $\rho^\beta(r) := \beta \log \mathbb{E}_{\omega \sim q} [\exp(-r \cdot \phi(\omega)/\beta)]$. The cost function $C(r) = \rho^\beta(-r)$ associated with this risk exactly corresponds to the* logarithmic market scoring rule *(LMSR). Its associated convex function $\alpha^\beta$ over distributions is the scaled relative entropy $\alpha^\beta(p) = \beta \, \mathrm{KL}(p \,|\, q)$. As discussed in [2, 13], the entropic risk is closely related to* exponential utility $U_\beta(w) := -\frac{1}{\beta} \exp(-\beta w)$. *Indeed, $\rho^\beta(r) = -U_\beta \left( \mathbb{E}_{\omega \sim q} [U_\beta(r \cdot \phi(\omega))] \right)$ which is just the negative certainty equivalent of the position $r$ — i.e., the amount of cash an agent with utility $U_\beta$ and belief $q$ would be willing to trade for the uncertain position $r$. Due to the monotonicity of $U_\beta^{-1}$, it follows that a trader maximizing expected utility $\mathbb{E}_{\omega \sim q} [U_\beta(r \cdot \phi(\omega))]$ of holding position $r$ is equivalent to minimizing the entropic risk $\rho^\beta(r)$.*

For technical reasons, in addition to the standard assumptions for convex risk measures, we will also make two weak regularity assumptions. These are similar to properties required of cost functions in the prediction market literature (cf. [1, Theorem 3.2]):

- *Expressiveness*: $\quad \rho$ is everywhere differentiable, and $\mathrm{closure}\{\nabla \rho(r) : r \in \mathcal{R}\} = \Pi$.
- *Strict risk aversion*: the *Convexity* inequality is strict unless $r - r' = c\,r^{\$}$ for some $c \in \mathbb{R}$.

As discussed in [1], expressiveness is related to the dual formulation given above; roughly, it says that the agent must take into account every possible expected value of the securities when calculating the risk. Strict risk aversion says that an agent should strictly prefer a mixture of positions, unless of course the difference is outcome-independent.

Under these assumptions, the representation result of Theorem 1 and a similar result for cost functions [1, Theorem 3.2]) coincide and we are able to show that cost functions and risk measures are exactly the same object; we write $\rho_C(r) = C(r)$ when we think of $C$ as a risk measure. Unfolding the definition of cost now using cash invariance, we have $\rho_C(s - dr + \mathrm{cost}(dr; s)r^{\$}) = \rho_C(s - dr) - \mathrm{cost}(dr; s) = C(s - dr) - C(s - dr) + C(s) = \rho_C(s)$. Thus, we may view a potential-based market maker as a *constant-risk* agent.

## 2.2 Trading Dynamics and Aggregation

As described above, we consider traders who approach the market maker sequentially and at random, and select the optimal trade based on their current position, the market state, and the cost function $C$.

As we just observed, we may think of the market maker as a constant-risk agent with $\rho_C = C$. Let us examine the optimization problem faced by the trader with position $r$ when the current market state is $s$. This trader will choose a portfolio $dr^*$ from the market maker so as to minimise her risk:

$$dr^* \in \underset{dr \in \mathbb{R}^k}{\arg\min}\, \rho\left(r + dr - \text{cost}(dr)r^\$\right) = \underset{dr \in \mathbb{R}^k}{\arg\min}\, \rho(r + dr) + \rho_C(s - dr)\,. \tag{1}$$

Since, by the cash invariance of $\rho$ and the definition of cost, the objective is $\rho(r + dr) + \rho_C(s - dr) - \rho_C(s)$, and $\rho_C(s)$ does not depend on $dr$. Thus, if we think of $F(r, s) = \rho(r) + \rho_C(s)$ as a kind of "social risk", we can define the *surplus* as simply the net risk taken away by an optimal trade, namely $F(r, s) - F(r + dr^*, s - dr^*)$.

We can now state our central question: *if a set of $N$ such traders arrive at random and execute optimal (or perhaps near-optimal) trades with the market maker, will the market state converge to the optimal risk, and if so how fast?* As discussed in the introduction, this is precisely the question asked in [2, 13] that we set out to answer. To do so we will draw a close connection to the literature on distributed optimization algorithms for machine learning. Specifically, if we encode the entire state of our system in the positions $R = (r_0 = s, r_1, \ldots, r_n)$ of the market maker and each of the $n$ traders, we may view the optimal trade in eq. (1) as performing a *coordinate descent* step, by optimizing only with respect to coordinates $0$ and $i$. We build on this connection in Section 4 and leverage a generalization of coordinate descent methods to show the following in Theorem 4: *If a set of risk-based traders is sampled at random to sequentially trade in the market, the market state and prices converge to within $\epsilon$ of the optimal total risk in $O(1/\epsilon)$ rounds.*

In fact, under mild smoothness assumptions on the cost potential function $C$, we can improve this rate to $O(\log(1/\epsilon))$. We can also relax the optimality of the trader behavior; as long as traders find a trade $dr$ which extracts at least a constant fraction of the surplus, the rate remains intact.

With convergence rates in hand, the next natural question might be: *to what* does the market converge? Abernethy et al. [2] show that when traders minimize expected exponential utility and have exponential family beliefs, the market equilibrium price can be thought of as a weighted average of the parameters of the traders, with the weights being a measure of their risk tolerance. Even though our setting is far more general than exponential utility and exponential families, the framework we develop can also be used to show that their results can be extended to interactions between traders who have what we call "compatible" risks and beliefs. Specifically, for any risk-based trader possessing a risk $\rho$ with dual $\alpha$, we can think of that trader's "belief" as the least surprising distribution $p$ according to $\alpha$. This view induces a family of distributions (which happen to be *generalized exponential families* [11]) that are parameterized by the initial positions of the traders. Furthermore, the risk tolerance $b$ is given by how sensitive this belief is to small changes of an agent's position. The results of [2] are then a special case of our Theorem 8 for agents with $\rho$ being entropic risk (cf. Example 1): *If each trader $i$ has risk tolerance $b_i$ and a belief parameterized by $\theta_i$, and the initial market state is $\theta_0$, then the equilibrium state of the market, to which the market converges, is given by $\theta^* = \frac{\theta_0 + \sum_i b_i \theta_i}{1 + \sum_i b_i}$ .*

As the focus of this paper is on the convergence, the details for this result are given in Appendix C.

The main insight that drives the above analysis of the interaction between a risk-based trader and a market maker is that each trade minimizes a global objective for the market that is the *infimal convolution* [6] of the traders' and market maker's risks. In fact, this observation naturally generalizes to trades between three or more agents and the same convergence analysis applies. In other words, our analysis also holds when bilateral trade with a fixed market maker is replaced by multilateral trade among arbitrarily overlapping subsets of agents. Viewed as a graph with agents as nodes, the standard prediction market framework is represented by the star graph, where the central market market interacts with traders sequentially and individually. More generally we have what we call a *trading network*, in which the structure of trades can form arbitrary connected graphs or even hypergraphs. An obvious choice is the complete graph, which can model a *decentralized* market, and in fact we can even compare the convergence rate of our dynamics between the centralized and decentralized models; see Appendix D.2 and the discussion in § 5.

## 3 General Trading Dynamics

The previous section described the two agent case of what is more generally known as the *optimal risk allocation problem* [6] where two or more agents express their preferences for positions via risk measures. This is formalized by considering $N$ agents with risk measures $\rho_i : \mathcal{R} \to \mathbb{R}$ for $i \in [N] := \{1, \ldots, N\}$ who are asked to split a position $r \in \mathcal{R}$ in to per-agent positions $r_i \in \mathcal{R}$ satisfying $\sum_i r_i = r$ so as to minimise the total risk $\sum_i \rho_i(r_i)$. They note that the value of the total risk is given by the *infimal convolution* $\wedge_i \rho_i$ of the individual agent risks — that is,

$$(\wedge_i \rho_i)(r) := \inf \left\{ \sum_i \rho_i(r_i) : \sum_i r_i = r \, , \, r_i \in \mathcal{R} \right\}. \tag{2}$$

A key property of the infimal convolution, which will underly much of our analysis, is that its convex conjugate is the sum of the conjugates of its constituent functions. See *e.g.* [23] for a proof.

$$(\wedge_i \rho_i)^* = \sum_{i \in [N]} \rho_i^* \, . \tag{3}$$

One can think of $\wedge_i \rho_i$ as the "market risk", which captures the risk of the entire market (*i.e.*, as if it were a single risk-based agent) as a function of the net position $\sum_i r_i$ of its constituents. By definition, eq. (2) says that the market is trying to reallocate the risk so as to minimize this net risk. This interpretation is confirmed by eq. (3) when we interpret the duals as penalty functions as above: the penalty of $\pi$ is the sum of the penalties of the market participants.

As alluded to above, we allow our agents to interact round by round by conducting trades, which are simply the exchange of outcome-contingent securities. Since by assumption our position space $\mathcal{R}$ is closed under linear combinations, a trade between two agents is simply a position which is added to one agent and subtracted from another. Generalizing from this two agent interaction, a trade among a set of agents $S \subseteq [N]$ is just a collection of trade vectors, one for each agent, which sum to 0. Formally, let $S \subseteq [N]$ be a subset of agents. A *trade on $S$* is then a vector of positions $dr \in \mathcal{R}^N$ (*i.e.*, a matrix in $\mathbb{R}^{N \times k}$) such that $\sum_{i \in S} dr_i = 0 \in \mathcal{R}$ and $dr_i = 0$ for all $i \notin S$. This last condition specifies that agents not in $S$ do not change their position.

A key quantity in our analysis is a measure of how much the total risk of a collection of traders drops due to trading. Given some subset of traders $S$, the *$S$-surplus* is a function $\Phi_S : \mathcal{R}^N \to \mathbb{R}$ defined by $\Phi_S(r) = \sum_{i \in S} \rho_i(r_i) - (\wedge_i \rho_i)(\sum_{i \in S} r_i)$ which measures the maximum achievable drop in risk (since $\wedge_i \rho_i$ is an infimum). In particular, $\Phi(r) := \Phi_{[N]}(r)$ is the *surplus function*. The trades that achieve this optimal drop in risk are called *efficient*: given current state $r \in \mathcal{R}^N$, a trade $dr \in \mathcal{R}^N$ on $S \subseteq [N]$ is *efficient* if $\Phi_S(r + dr) = 0$.

Our following key result shows that efficient trades have remarkable structure: once the state $r$ and subset $S$ is specified, there is a unique efficient trade, up to cash transfers. In other words, the surplus is removed from the position vectors and then redistributed as cash to the traders; the choice of trade is merely in how this redistribution takes place. The fact that the derivatives match has strong intuition from prediction markets: agents must agree on the price.[2] The proof is in Appendix A.1.

**Theorem 2.** *Let $r \in \mathcal{R}^N$ and $S \subseteq [N]$ be given.*

  i. *The surplus is always finite: $0 \leq \Phi_S(r) < \infty$.*

 ii. *The set of efficient trades on $S$ is nonempty.*

iii. *Efficient trades are unique up to zero-sum cash transfers: Given efficient trades $dr^*, dr \in \mathcal{R}^N$ on $S$, we have $dr = dr^* + (z_1 r^\$, \ldots, z_N r^\$)$ for some $z \in \mathbb{R}^N$ with $\sum_i z_i = 0$.*

 iv. *Traders agree on "prices": A trade $dr$ on $S$ is efficient if and only if for all $i, j \in S$, $\nabla \rho_i(r_i + dr_i) = \nabla \rho_j(r_j + dr_j)$.*

  v. *There is a unique "efficient price": If $dr$ is an efficient trade on $S$, for all $i \in S$ we have $\nabla \rho_i(r_i + dr_i) = -\pi_S^*$, where $\pi_S^* = \arg\min_{\pi \in \Pi} \sum_{i \in S} \alpha_i(\pi) - \langle \pi, \sum_{i \in S} r_i \rangle$.*

The above properties of efficient trades drive the remainder of our convergence analysis of network dynamics. It also allows us to write a simple closed form for the market price when traders share a common risk profile (Theorem 8). Details are in Appendix C. Beyond our current focus on rates, Theorem 2 has implications for a variety of other economic properties of trade networks. For example, in Appendix B we show that efficient trades correspond to fixed points for more general dynamics, market clearing equilibria, and equilibria of natural bargaining games among the traders.

Recall that in the prediction market framework of [13], each round has a single trader, say $i > 1$, interacting with the market maker who we will assume has index 1. In the notation just defined this corresponds to choosing $S = \{1, i\}$. We now wish to consider richer dynamics where groups of two or more agents trade efficiently each round. To this end will we call a collection $\mathcal{S} = \{S_j \subseteq [N]\}_{j=1}^m$ of groups of traders a *trading network* and assume there is some fixed distribution $D$ over $\mathcal{S}$ with full support. A *trade dynamic* over $\mathcal{S}$ is a process that begins at $t = 0$ with some initial positions $r^0 \in \mathcal{R}^N$ for the $N$ traders, and at each round $t$, draws a random group of traders $S^t \in \mathcal{S}$ according to $D$, selects some efficient trade $dr^t$ on $S$, then updates the trader positions using $r^{t+1} = r^t + dr^t$.

For the purposes of proving the convergence of trade dynamics, a crucial property is whether all traders can directly or indirectly affect the others. To capture this we will say a trade network is *connected* if the hypergraph on $[N]$ with edges given by $\mathcal{S}$ is connected; *i.e.*, information can propagate throughout the entire network. Dynamics over classical prediction markets are always connected since any pair of groups from its network will always contain the market maker.

## 4 Convergence Analysis of Randomized Subspace Descent

Before briefly reviewing the literature on coordinate descent, let us see why this might be a useful way to think of our dynamics. Recall that we have a set $\mathcal{S}$ of subsets of agents, and that in each step, an efficient trade $dr$ is chosen which only modifies the positions of agents in the sampled $S \in \mathcal{S}$. Thinking of $(r_1, \ldots, r_N)$ as a vector of dimension $N \cdot k$ vector (recall $\mathcal{R} = \mathbb{R}^k$), changing $r^t$ to $r^{t+1} = r^t + dr$ thus only modifies $|S|$ blocks of $k$ entries. Moreover, efficiency ensures that $dr$ *minimizes* the sum of the risks of agents in $S$. Hence, ignoring for now the constraint that the sum of the positions must remain constant, the trade dynamic seems to be performing a kind of block coordinate descent of the surplus function $\Phi$.

### 4.1 Randomized Subspace Descent

Several randomized coordinate descent methods have appeared in the literature recently, with increasing levels of sophistication. While earlier methods focused on updates which only modified disjoint blocks of coordinates [18, 22], more recent methods allow for more general configurations, such as overlapping blocks [17, 16, 20]. In fact, these last three methods are closest to what we study here; the authors consider an objective which decomposes as the sum of convex functions on each coordinate, and study coordinate updates which follow a graph structure, all under the constraint that coordinates sum to 0. Despite the similarity of these methods to our trade dynamics, we require even more general updates, as we allow coordinate $i$ to correspond to arbitrary subsets $S_i \in \mathcal{S}$.

Instead, we establish a unification of these methods which we call *randomized subspace descent (RSD)*, listed in Algorithm 1. Rather than blocks of coordinates or specific linear constraints, RSD abstracts away these constructs by simply specifying "coordinate subspaces" in which the optimization is to be performed. Specifically, the algorithm takes a list of projection matrices $\{\Pi_i\}_{i=1}^n$ which define the subspaces, and at each step $t$ selects a $\Pi_i$ at random and tries to optimize the objective under the constraint that it may only move within the image space of $\Pi_i$; that is, if the current point is $x^t$, then $x^{t+1} - x^t \in \text{im}(\Pi_i)$.

Before stating our convergence results for Algorithm 1, we will need a notion of smoothness relative to our subspaces. Specifically, we say $F$ is $L_i$-$\Pi_i$-*smooth* if for all $i$ there are constants $L_i > 0$ such that for all $y \in \text{im}(\Pi_i)$,

$$F(x + y) \le F(x) + \langle \nabla F(x), y \rangle + \tfrac{L_i}{2} \|y\|_2^2. \tag{4}$$

Finally, let $F^{\min} := \min_{y \in \text{span}\{\text{im}(\Pi_i)\}_i} F(x^0 + y)$ be the global minimizer of $F$ subject to the constraints from the $\Pi_i$. Then we have the following result for a constant $\mathcal{R}(x^0)$ which increases in:

---

**ALGORITHM 1:** Randomized Subspace Descent

---

**Input**: Smooth convex function $F : \mathbb{R}^n \to \mathbb{R}$, initial point $x^0 \in \mathbb{R}^n$, matrices $\{\Pi_i \in \mathbb{R}^{n \times n}\}_{i=1}^m$,
       smoothness parameters $\{L_i\}_{i=1}^m$, distribution $p \in \Delta_m$

  **for** iteration $t$ in $\{0, 1, 2, \cdots\}$ **do**
      sample $i$ from $p$
      $x^{t+1} \leftarrow x^t - \frac{1}{L_i} \Pi_i \nabla F(x^t)$

**end**

---

(1) the distance from the point $x^0$ to furthest minimizer of $F$, (2) the Lipschitz constants of $F$ w.r.t. the $\Pi_i$, and (3) the connectivity of the hypergraph induced by the projections.

**Theorem 3.** *Let $F$, $\{\Pi_i\}_i$, $\{L_i\}_i$, $x^0$, and $p$ be given as in Algorithm 1, with the condition that $F$ is $L_i$-$\Pi_i$-smooth for all $i$. Then $\mathbb{E}\left[F(x^t) - F^{\min}\right] \leq 2\mathcal{R}^2(x^0)/t$.*

The proof is in Appendix D. Additionally, when $F$ is strongly convex, meaning it has a uniform local quadratic lower bound, RSD enjoys faster, linear convergence. Formally, this condition requires $F$ to be $\mu$-strongly convex for some constant $\mu > 0$, that is, for all $x, y \in \operatorname{dom} F$ we require

$$F(y) \geq F(x) + \nabla F(x) \cdot (y - x) + \tfrac{\mu}{2}\|y - x\|^2 . \tag{5}$$

The statement and details of this stronger result is given in Appendix D.1.

Importantly for our setting these results only track the progress per iteration. Thus, they apply to more sophisticated update steps than a simple gradient step as long as they improve the objective by at least as much. For example, if in each step the algorithm computed the exact minimizer $x^{t+1} = \arg\min_{y \in \operatorname{im}(\Pi_i)} F(x^t + y)$, both theorems would still hold.

## 4.2 Convergence Rates for Trade Dynamics

To apply Theorem 3 to the convergence of trading dynamics, we let $F = \Phi$ and $x = (r_1, \ldots, r_N) \in \mathcal{R}^N \cong \mathbb{R}^{Nk}$ be the joint position of all agents. For each subset $S \in \mathcal{S}$ of agents, we have a subspace of $\mathcal{R}^N$ consisting of all possible trades on $S$, namely $\{dr \in \mathcal{R}^N : dr_i = 0 \text{ for } i \neq S, \sum_{i \in S} dr_i = 0\}$, with corresponding projection matrix $\Pi_S$. For the special case of prediction markets with a centralized market maker, we have $N - 1$ subspaces $\mathcal{S} = \{\{1, i\} : i \in \{2, \ldots, N\}\}$ and $\Pi_{1,i}$ projects onto $\{dr \in \mathcal{R}^N : dr_i = -dr_1, dr_j = 0 \text{ for } j \neq 1, i\}$. The intuition of coordinate descent is clear now: the subset $S$ of agents seek to minimize the total surplus within the subspace of trades on $S$, and thus the coordinate descent steps of Algorithm 1 will correspond to roughly efficient trades.

We now apply Theorem 3 to show that trade dynamics achieve surplus $\epsilon > 0$ in time $O(1/\epsilon)$. Note that we will have to assume the risk measure $\rho_i$ of agent $i$ is $L_i$-smooth for some $L_i > 0$. This is a very loose restriction, as our risk measures are all differentiable by the expressiveness condition.

**Theorem 4.** *Let $\rho_i$ be an $L_i$-smooth risk measure for all $i$. Then for any connected trade dynamic, we have $\mathbb{E}\left[\Phi(r^t)\right] = O(1/t)$.*

*Proof.* Taking $L_S = \max_{i \in S} L_i$, one can check that $F$ is $L_S$-$\Pi_S$-smooth for all $S \in \mathcal{S}$ by eq. (4). Since Algorithm 1 has no state aside from $x^t$, and the proof of Theorem 3 depends only the drop in $F$ per step, any algorithm selecting the sets $S \in \mathcal{S}$ with the same distribution and satisfying $F(x^{t+1}) \leq F(x^t - \frac{1}{L_i}\Pi_i \nabla F(x^t))$ will yield the same convergence rate. As trade dynamics satisfy $F(x^{t+1}) = \min_{y \in \mathbb{R}^{Nk}} F(x^t - \Pi_i y)$, this property trivially holds, and so Theorem 3 applies. $\square$

If we assume slightly more, that our risk measures have local quadratic lower bounds, then we can obtain linear convergence. Note that this is also a relatively weak assumption, and holds whenever the risk measure has a Hessian with only one zero eigenvalue (for $r^\$$) at each point. This is satisfied, for example, by all the variants of entropic risk we discuss in the paper. The proof is in Appendix D.

**Theorem 5.** *Suppose for each $i$ we have a continuous function $\mu_i : \mathcal{R} \to \mathbb{R}_+$ such that for all $r$, risk $\rho_i$ is $\mu_i(r)$-strongly convex with respect to $r^{\$\perp}$ in a neighborhood of $r$; in other words, eq. (5) holds for $F = \rho_i$, $\mu = \mu_i(r)$, and all $y$ in a neighborhood of $r$ such that $(r - y) \cdot r^\$ = 0$. Then for all connected trade dynamics, $\mathbb{E}\left[\Phi(r^t)\right] = O(2^{-t})$.*

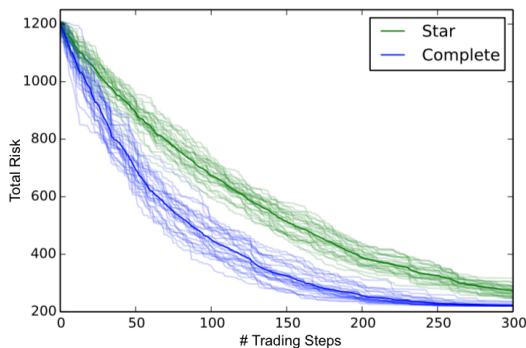

| Graph | $\|V(G)\|$ | $\|E(G)\|$ | $\lambda_2(G)$ |
|---|---|---|---|
| $K_n$ | $n$ | $n(n-1)/2$ | $n$ |
| $P_n$ | $n$ | $n-1$ | $2(1-\cos\frac{\pi}{n})$ |
| $C_n$ | $n$ | $n$ | $2(1-\cos\frac{2\pi}{n})$ |
| $K_{\ell,k}$ | $\ell+k$ | $\ell k$ | $k$ |
| $B_k$ | $2^k$ | $k2^{k-1}$ | $2$ |

Table 1: Algebraic connectivities for common graphs.

Figure 1: Average (in bold) of 30 market simulations for the complete and star graphs. The empirical gap in iteration complexity is just under 2 (cf. Fig. 3).

Amazingly, the convergence rates in Theorem 4 and Theorem 5 hold for *all* connected trade dynamics. The constant hidden in the $O(\cdot)$ does depend on the structure of the network but can be explicitly determined in terms its *algebraic connectivity*. This is discussed further in Appendix D.2.

The intuition behind these convergence rates given here is that agents in whichever group $S$ is chosen always trade to fully minimize their surplus. Because the proofs (in Appendix D) of these methods merely track the reduction in surplus per trading round, the bounds apply as long as the update is at least as good as a gradient step. In fact, we can say even more: if only an $\epsilon$ fraction of the surplus is taken at each round, the rates are still $O(1/(\epsilon t))$ and $O((1-\epsilon\mu)^t)$, respectively. This suggests that our convergence results are robust with respect to the model of rationality one employs; if agents have bounded rationality and can only compute positions which approximately minimize their risk, the rates remain intact (up to constant factors) as long as the inefficiency is bounded.

# 5  Conclusions & Future Work

Using the tools of convex analysis to analyse the behavior of markets allows us to make precise, quantitative statements about their global behavior. In this paper we have seen that, with appropriate assumptions on trader behaviour, we can determine the rate at which the market will converge to equilibrium prices, thereby closing some open questions raised in [2] and [13].

In addition, our newly proposed trading networks model allow us to consider a variety of prediction market structures. As discussed in §3, the usual prediction market setting is *centralized*, and corresponds to a star graph with the market maker at the center. A *decentralized* market where any trader can trade with any other corresponds to a complete graph over the traders. We can also model more exotic networks, such as two or more market maker-based prediction markets with a risk minimizing arbitrageur or small-world networks where agents only trade with a limited number of "neighbours".

Furthermore, because these arrangements are all instances of trade networks, we can immediately compare the convergence rates across various constraints on how traders may interact. For example, in Appendix D.2, we show that a market that trades through a centralized market maker incurs an quantifiable efficiency overhead: convergence takes twice as long (see Figure 1). More generally, we show that the rates scale as $\lambda_2(G)/|E(G)|$, allowing us to make similar comparisons between arbitrary networks; see Table 1. This raises an interesting question for future work: given some constraints such as a bound on how many traders a single agent can trade with, the total number of edges, etc, which network optimizes the convergence rate of the market? These new models and the analysis of their convergence may provide new principles for building and analyzing distributed systems of heterogeneous and self-interested learning agents.

### Acknowledgments

We would like to thank Matus Telgarsky for his generous help, as well as the lively discussions with, and helpful comments of, Sébastien Lahaie, Miro Dudík, Jenn Wortman Vaughan, Yiling Chen, David Parkes, and Nageeb Ali. MDR is supported by an ARC Discovery Early Career Research Award (DE130101605). Part of this work was developed while he was visiting Microsoft Research.

## Footnotes

[1] It is more common in the prediction market literature for $s$ to be a *liability* vector, tracking what the market maker stands to lose instead of gain. Here we adopt positive positions to match the convention for risk measures.

[2]As intuition for the term "price", consider that the highest price-per-unit agent $i$ would be willing to pay for an infinitesimal quantity of a position $dr_i$ is $dr_i \cdot (-\nabla \rho_i(r_i))$, and likewise the lowest price-per-unit to sell. Thus, the entries of $-\nabla \rho_i(r_i)$ act as the "fair" prices for their corresponding basis positions/securities.

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
