[Supplementary Material]

# A Proofs of Results

## A.1 Proof of Theorem 2

*Proof.* We begin with (iv). By [12, eq. X1.3.4.5], which gives a very general result about infimal convolutions, we have that the condition $\Phi_S(r + dr) = 0$ implies the existence of some $\pi$ such that $\nabla\rho_i(r_i + dr_i) = \pi$ for all $i \in S$. Conversely, we can appeal to [12, Prop X1.3.4.2], which again is for general infimal convolutions, to conclude $\Phi(r + dr) = 0$.

Now for $y \in \mathcal{R}$ consider $F(y) := (\wedge_{i \in S}\rho_i)(y)$, and let $x = \sum_{i \in S} r_i$. By eq. (3) and the definition of the conjugate, we have $\pi = -\nabla F(x) = \pi_S^*$ as defined in (v). Turning to (i), note that as $\text{relint}\Pi \subseteq \text{dom }\rho_i^*$ for all $i$ by the expressiveness condition, we have $\text{relint}\Pi \subseteq \cap_{i \in S} \text{dom }\rho_i^* = \text{dom }F^*$. Now [12, Prop X1.3.4.1, eq. (XI.3.4.2)] gives us some $r' \in \mathcal{R}^N$ with $\sum_{i \in S} r_i' = x$, and $r_i' = r_i$ for $i \notin S$, such that $F(x) = \sum_{i \in S} \rho_i(r_i')$, giving us $\Phi_S(r) = \sum_{i \in S}\rho_i(r_i) - F(x) = \sum_{i \in S}\rho_i(r_i) - \sum_{i \in S}\rho_i(r_i') < \infty$. As $\Phi_S(r) \geq 0$ by definition, we have (i), and taking $dr = r' - r$, which is a trade as $\sum_i dr_i = \sum_{i \in S} r' - \sum_{i \in S} r_i + \sum_{i \notin S}(r_i' - r_i) = x - x + 0$, we also have (ii).

We can now settle (iii): by cash invariance, it is clear that $\nabla\rho_i(r_i + dr_i) = \nabla\rho_i(r_i + dr_i + z_i r^\$)$ for all $z_i \in \mathbb{R}$, and the strict risk aversion property says that these are the only such positions with the same derivative (otherwise convexity would imply $\rho_i$ is flat in between, a contradiction). The requirement that $\sum_i dr_i = 0$ ensures $\sum_i z_i = 0$. $\qquad\square$

## A.2 Proof of Theorem 8

*Proof.* Working with compressed positions, we have $\rho_i(r_i) = b\rho(\theta_i + r_i/b) - b\rho(\theta_i)$, where we have overloaded $\rho(r_i) = \rho(X[r_i])$. Taking $r_i^0 = 0$ for all $i$, by Theorem 7 and Theorem 2, the market clearing price is the unique price such that $\nabla\rho_i(dr_i) = \pi^*$ for all $i$, for some trade $dr \in \mathcal{R}^N$. By definition of $\rho_i$ we have $\nabla\rho_i(dr_i) = \nabla\rho(\theta_i + dr_i/b_i)$. Now letting $\bar{b} = \sum_{j=1}^N b_j$, we see that taking $dr_i = b_i \sum_{j=1}^N b_j\theta_j/\bar{b} - b_i\theta_i$ gives us a valid trade with $\sum_i dr_i = \bar{b}\sum_{j=1}^N b_j\theta_j/\bar{b} - \sum_{i=1}^N b_i\theta_i = 0$, and by symmetry we have $\pi^* = \nabla\rho(\sum_{j=1}^N b_j\theta_j/\bar{b})$, meaning $\theta^* = \sum_{j=1}^N b_j\theta_j/\bar{b}$ as desired. $\qquad\square$

# B Bargaining and Equilibria in Trade Networks

The key result of Theorem 2 concerning efficient trades provide a wealth of structure to help us understand market behaviour. The next result shows that efficient trades are fixed points of a dynamic. We say a state $r \in \mathcal{R}^N$ is a *fixed point* of a dynamic $D$ if $r^t = r$ implies $r^s = r$ for all $s > t$.

**Theorem 6.** *Let $D(\mathcal{S}, p)$ be a connected trade dynamic. Then $r \in \mathcal{R}^N$ is a fixed point of $D$ if and only if it is efficient.*

*Proof.* If $\Phi(r) = 0$, then $dr = 0$ is the only efficient monotone trade, so $r$ is a fixed point of $D$. Conversely, if $r$ is a fixed point of $D$, then as $p$ has full support, we must have $\Phi_S(r) = 0$ for all $S \in \mathcal{S}$; otherwise with constant probability we would have $S = S^t$ for some $t$ and thus some efficient trade $dr^t \neq 0$ and $r^{t+1} \neq r$. By Theorem 2 (iv), this means that for all $S \in \mathcal{S}$ and all $i, j \in S$ we have $\nabla\rho_i(r_i) = \nabla\rho_j(r_j)$. This gives us an equivalence classes of derivatives for each $S$, and by connectedness $\mathcal{S}$, the equivalence classes in fact must coincide. Thus, we have some $\pi$ for which $\nabla\rho_j(r_j) = \pi$ for all $j \in [N]$, and again by Theorem 2 (iv) we conclude $\Phi(r) = 0$. $\qquad\square$

We observe that the combination of Theorems 2 and 6 means that for connected trade dynamics $D$, the fixed point of $D$ corresponds to the efficient price $\pi^*$ and hence the efficient trade for the overall market.

The result of Theorem 6 is somewhat surprising — not only is there a unique equilibrium (up to cash transfers) for all connected dynamics, but all connected dynamics have the *same* equilibrium! If one restricts to connected graphical networks, this means that the equilibrium does not depend on the network structure. The power of our framework is that the equilibrium analysis holds regardless of the way agents interact, as long as information is allowed to spread to all agents eventually. In fact, one could even consider an arbitrary process choosing subsets $S^t$ of agents to trade at each time $t$; if

the set $\mathcal{S}$ of subsets which are visited infinitely often yields a connected hypergraph, then the proof Theorem 6 still applies.

We now explore two other types of equilibria in this setting. The first is a natural game-theoretic equilibrium where no group of agents can trade for mutual gain. The second, in the following subsection, looks at the classical market clearing condition in this setting.

**Proposition 1.** *Given connected $\mathcal{S}$, a point $r$ is a game-theoretic equilibrium, meaning there is no set of agents $S \in \mathcal{S}$ which can mutually benefit by trading, if and only if $r$ is efficient.*

*Proof.* Following the same logic as the proof of Theorem 6, we first argue that $\Phi_S(r) = 0$ for all $S$, and by connectedness this gives us some $\pi$ such that $\nabla \rho_i(r) = -\pi$ for all $i$, from which we conclude $\Phi(r) = 0$. $\qquad\square$

## B.1 Market clearing

Viewing the basis positions $\phi(\cdot)_i$ as goods in a marketplace, and the initial (compact) positions $r_i^0 \in \mathcal{R}$ as the endowments of the agents, it is natural to ask if there is a set of prices for the goods which clears the market. That is, if we set some price vector $\pi \in \Pi$ such that each agent buys some vector $dr_i$ at those prices, we wish to find $\pi$ such that $\sum_i dr_i = 0$.

Let us examine the optimization problem for each agent. Upon purchasing a bundle of goods $dr_i \in \mathcal{R}$ at prices $\pi$, for a total cost of $dr_i \cdot \pi$, the agent has net position $r_i^0 + dr_i - (dr_i \cdot \pi) r^\$$. In particular, it is now clear that we must have $r^\$ \cdot \pi = 1$; otherwise, if e.g. $r^\$ \cdot \pi < 1$, an agent purchasing $\lambda r^\$$ at cost would have final position $\lambda(1 - r^\$ \cdot \pi) r^\$$, which corresponds to an arbitrarily large risk-free payoff. Finally, as we simply factor the price of a bundle back into the position itself, the agent's purchasing decision is equivalent to choosing some $dr_i$ such that $dr_i \cdot \pi = 0$.

As the utility of the agent is entirely captured by its risk measure $\rho_i$, we can clearly state the agent's optimization problem: given price vector $\pi$, agent $i$ chooses a bundle of goods $dr_i$ given by

$$dr_i \in \arg\min_{x : x \cdot \pi = 0} \rho_i(r_i^0 + x) . \tag{6}$$

Via the method of Lagrange multipliers, a minimizing bundle $dr_i$ in (6) must satisfy the constraint $\nabla \rho_i(r_i^0 + dr_i) = \lambda \pi$ for some $\lambda \in \mathbb{R}$. By cash invariance, we know that for all $\pi' \in -\nabla \rho_i(\cdot)$ we have $r^\$ \cdot \pi' = 1$, and as $r^\$ \cdot \pi = 1$ by the above argument, we must have $\lambda = 1$. We now have $\nabla \rho_i(r_i^0 + dr_i) = -\pi$ for all $i$. If we assume that $\pi$ clears the market, then we additionally have $\sum_i dr_i = 0$, meaning $dr$ is a valid trade, and thus Theorem 2 implies $\Phi(r^0 + dr) = 0$. Again by Theorem 2, we conclude that in fact $\pi = \pi^* = \arg\min_{\pi \in \Pi} \sum_i \alpha_i(\pi) - \langle \pi, \sum_i r_i^0 \rangle$.

Of course, it remains to show that the market clearing condition can be satisfied given the constraint $dr_i \cdot \pi^* = 0$. By Theorem 2 (ii), we know there exists an efficient trade $dr'$, and by (iii) it is unique up to zero-sum cash transfers. Thus, taking $dr_i = dr_i' - (dr_i' \cdot \pi^*) r^\$$, which is zero-sum as $\sum_i dr_i' = 0$, we see that there is a unique optimal allocation $r = r^0 + dr$ to the agents, and which clears the market.

From the above discussion, we have the following result which refines the equilibrium concepts above; instead of a unique equilibrium up to zero-sum cash transfers, the market clearing allocation fixes a single position for each agent.

**Theorem 7.** *Let goods $\{\phi(\cdot)_i\}_{i=1}^k$, initial endowments $r^0 \in \mathcal{R}^N$, and (negative) utilities $\{\rho_i\}_{i=1}^N$ be given. Then the unique market clearing price $\pi^*$ is given by Theorem 2 (v) and yields a unique and efficient allocation $dr$, which satisfies $dr_i \cdot \pi^* = 0$ for all $i$.*

Intuitively, the market clearing allocation gives us a benchmark for the proper way to "divide up the surplus" among the agents; given an efficient trade $dr$, perhaps the outcome of bargaining, each agent $i$ should readjust by subtracting $dr \cdot \pi^*$ in cash.

## C  Optimal Allocation for Risk Compatible Traders

The price matching property of efficient trades (Theorem 2) allows us to find closed form solutions for the optimal allocation of risk in a market when the traders' risks are suitably "compatible".

These risk measures are derived from a *base* risk $\rho$ which allows us to capture a notion of beliefs and varying levels of risk aversion, as follows. Let

$$\rho_{b,s}(r) := b\rho(s + r/b) - b\rho(s), \tag{7}$$

where $b > 0$ is the *risk tolerance* and $s \in \mathcal{R}$ is some reference position. We will call the risk measures $\rho_{b,s}$ *compatible* with $\rho$ and can view them as a combination of a translation $\rho(r) = \rho(s + r) - \rho(s)$, which simulates the additional risk imposed by $r$ having already taken on position $s$, and a *perspective transform* $\rho(r) = b\rho(r/b)$, where a higher $b$ corresponds to a less risk-averse agent.[3]

In general, for an agent with risk measure $\rho : \mathcal{R} \to \mathbb{R}$, we can think of $-\nabla\rho(0)$ as the agent's belief. This can be justified in two ways: (1) when $\rho$ is the (negative) certainty equivalent for exponential utility, for an agent with belief $p_s$, then $-\nabla\rho(0) = p_s$; (2) the penalty function $\alpha$, which specifies how "surprising" each distribution $p \in \Delta_\Omega$ is, is minimized at $p = -\nabla\rho(0)$. For this reason, and the fact that $\nabla\rho_{b,s}(0) = \nabla\rho(s) = -p_s$, we may think of $\rho_{b,s}$ as the risk of an agent with initial belief $p_s$ and risk tolerance $b$.

Consider agents with inital beliefs $p_{s_i}$ and risk tolerances $b_i > 0$, meaning $\rho_i = \rho_{b_i,s_i}$. What is the final market price in terms of these parameters? Motivated by the prediction market setting, we would in particular like to see some sort of sensible *aggregation* of these beliefs. As we now show, the market clearing price corresponds to a risk-tolerance-weighted average of the agents' parameters.

**Theorem 8.** *Let $\rho$ be a risk measure, and let each agent $i$ have compatible risk measures $\rho_{b_i,s_i}$ initial belief $p_{s_i}$ and risk tolerance parameter $b_i > 0$. Then the market clearing price is given by $p_{s^*}$, where*

$$s^* = \frac{\sum_i b_i s_i}{\sum_i b_i} . \tag{8}$$

This result generalizes those in §5 of [2], where traders are assumed to maximize an expected utility of the form $U_b(w) = -b\exp(-w/b)$ under beliefs drawn from an exponential family with sufficient statistic given by the securities $\phi$. The above result shows that exactly the same weighted distribution of positions at equilibrium occurs for *any* family of risk-based agents, not just those derived from exponential utility via certainty equivalents [4]. In addition, this generalization shows that the agents need not have exponential family beliefs: their positions $\theta_i$ act as general natural parameters, and $1/b_i$ acts as a general measure of risk aversion. Finally, applying Theorem 4 or 5 establishes convergence rates for this setting, which addresses the future work in [2].

## D   Random Subspace Descent

For the analysis, we introduce a seminorm $\|\cdot\|_A$ which will measure the progress per iteration, and its dual $\|\cdot\|_A^*$.

$$\|x\|_A := \left( \sum_{i=1}^m \frac{p_i}{L_i} \|\Pi_i x\|_2^2 \right)^{1/2} . \tag{9}$$

$$\|y\|_A^* := \begin{cases} \langle A^+ y, y \rangle^{1/2} & \text{if } y \in \text{im}(A) \\ \infty & \text{otherwise.} \end{cases} \tag{10}$$

Note that $\|\cdot\|_A$ is a Euclidean seminorm $\|x\|_A = \langle Ax, x \rangle$ with $A = \sum_i \frac{p_i}{L_i} \Pi_i$. One can check that $\|\cdot\|_A^*$ is indeed the dual norm of $\|\cdot\|_A$, in the sense that $\left(\frac{1}{2}\|\cdot\|_A^2\right)^* = \frac{1}{2}\|\cdot\|_A^{*\,2}$, where the first $*$ denotes the convex conjugate.

Define $X(A) = \{x^0 + y : y \in \text{span}\{\text{im}(\Pi_i)\}_i\}$ to be the optimization domain, so that $F^{\min} = \min_{x \in X(A)} F(x)$, and let $F^{\text{arg}} := \arg\min_{x \in X(A)} F(x)$ denote the minimizers of $F$. Then we define the constant $\mathcal{R}(x^0)$ by

$$\mathcal{R}(x^0) := \max_{x \in X(A):F(x) \le F(x^0)} \max_{x^* \in F^{\text{arg}}} \|x - x^*\|_A^* , \tag{11}$$

the maximum distance, according to $\| \cdot \|_A^*$, between any minimizer of $F$ and any feasible point at least as good as $x^0$.

We now have the foundation to prove Theorem 3.

*Proof of Theorem 3.* To begin, suppose subspace $i$ is chosen at step $t$, and consider the update $x^{t+1} = x^t - y$ for $y \in \mathrm{im}(\Pi_i)$. The drop in the objective can be bounded using eq. (4),

$$F(x^t) - F(x^t - y) \geq \langle \nabla F(x^t), y \rangle - \frac{L_i}{2}\|y\|_2^2 \,. \tag{12}$$

By properties of orthogonal projections, we have

$$\arg\max_{y \in \mathrm{im}(\Pi_i)} \langle \nabla F(x^t), y \rangle - \frac{L_i}{2}\|y\|_2^2 \;=\; \arg\min_{y \in \mathrm{im}(\Pi_i)} \left\| y - \tfrac{1}{L_i}\nabla F(x^t) \right\|_2 \;=\; \tfrac{1}{L_i}\Pi_i \nabla F(x^t) \,,$$

and choice of $y$ gives our update in Algorithm 1. Substituting this $y$ into eq. (12) gives

$$F(x^t) - F(x^{t+1}) \geq \left\langle \nabla F(x^t), \tfrac{1}{L_i}\Pi_i \nabla F(x^t) \right\rangle - \frac{L_i}{2}\|\tfrac{1}{L_i}\Pi_i \nabla F(x^t)\|_2^2$$

$$= \frac{1}{2L_i}\|\Pi_i \nabla F(x^t)\|_2^2 \,.$$

Now looking at the expected drop in the objective, we have

$$F(x^t) - \mathbb{E}\left[ F(x^{t+1})|x^t \right] \geq \sum_{i=1}^m p_i \frac{1}{2L_i}\|\Pi_i \nabla F(x^t)\|_2^2 = \frac{1}{2}\|\nabla F(x^t)\|_A^2 \,. \tag{13}$$

To relate our per-round progress to the gap remaining, we observe that

$$F(x^t) - F^{\min} \leq \max_{x^* \in \arg\min_x F(x)} \langle \nabla F(x^t), x^* - x^t \rangle$$

$$\leq \max_{x^* \in \arg\min_x F(x)} \|\nabla F(x^t)\|_A \|x^* - x^t\|_A^*$$

$$\leq \|\nabla F(x^t)\|_A \max_{x^* \in \arg\min F} \max_{x:F(x) \leq F(x^0)} \|x^* - x\|_A^*$$

$$= \|\nabla F(x^t)\|_A \, \mathcal{R}(x^0) \,,$$

where we used convexity of $F$, the definition of the dual norm, the fact that $F(x^t)$ is non-increasing in $t$, and finally the definition of $\mathcal{R}$. We now have $F(x^t) - \mathbb{E}\left[ F(x^{t+1})|x^t \right] \geq (F(x^t) - F^{\min})^2/(2\mathcal{R}^2(x^0))$. The remainder of the proof follows an argument of [16] by analyzing $\Delta_t = \mathbb{E}\left[ F(x^t) - F^{\min} \right]$. From the last inequality we have $\Delta_{t+1} \leq \Delta_t - \Delta_t^2/2\mathcal{R}^2(x^0)$, and since $0 \leq \Delta_{t+1} \leq \Delta_t$, dividing by $\Delta_t \Delta_{t+1}$ gives $\Delta_t^{-1} \leq \Delta_{t+1}^{-1} - (2\mathcal{R}^2(x^0))^{-1}$. Summing these inequalities gives the result. $\qquad\square$

## D.1 Faster Rates for RSD

**Theorem 9.** *Let $F$, $\{\Pi_i\}_i$, $\{L_i\}_i$, $x^0$, and $p$ be given as in Algorithm 1, with the condition that $F$ is $L_i$-$\Pi_i$-smooth for all $i$, and additionally that $F$ is $\mu$-strongly convex with respect to $\| \cdot \|_A^*$. Then we have $\mathbb{E}\left[ F(x^t) - F^{\min} \right] \leq (1-\mu)^t (F(x^0) - F^{\min})$.*

*Proof of Theorem 9.* Our proof is essentially that of Nesterov [18, Thm 2] and Richtárik and Takáč [22, Thm 12]. By definition of $\mu$-strongly convex, we have for all $y \in \mathbb{R}^n$,

$$F(y) - F(x^t) \geq \langle \nabla F(x^t), y - x^t \rangle + \frac{\mu}{2}\|y - x^t\|_A^{*\,2} \,.$$

Independently minimizing each side of this inequality over $y$, we obtain from [22, Lemma 10],

$$F^{\min} - F(x^t) \geq -\tfrac{1}{2\mu}\|\nabla F(x^t)\|_A^2.$$

Now combining with eq. (13), we have

$$F(x^t) - \mathbb{E}\left[ F(x^{t+1})|x^t \right] \geq \frac{1}{2}\|\nabla F(x^t)\|_A^2 \geq \mu(F(x^t) - F^{\min}) \,.$$

Taking expectations and rearranging, we have $\mathbb{E}\left[ F(x^{t+1}) - F^{\min} \right] \leq (1-\mu)\mathbb{E}\left[ F(x^t) - F^{\min} \right]$, from which the result follows by induction. $\qquad\square$

## D.2 RSD for Graphs and Hypergraphs

In this section we consider a special case of Algorithm 1, where we have a linear constraint $\sum_i x_i = c$ on the coordinates, and the subspaces correspond to graphs (overlapping pairs of coordinates), or hypergraphs (overlapping subsets of coordinates).[4] In the graphical case, we will leverage existing results in spectral graph theory to analyze new graphs currently not considered in the literature. Note that we focus here on uniform probabilities to highlight the connections to spectral graph theory; for an analysis of the optimal probabilities, see Necoara et al. [16].

Let us first consider an optimization problem on the complete graph, which picks an edge $(i, j)$ uniformly at random and optimizes in coordinates $i$ and $j$ under the constraint that $x_i^{t+1} + x_j^{t+1} = x_i^t + x_j^t$. One can check that this corresponds to the projection matrix $\Pi_{(i,j)} = \frac{1}{2}(e_i - e_j)(e_i - e_j)^\top$, where $e_i$ is the $i$th standard unit vector. Assuming a global smoothness constant $L$, one can calculate

$$A = \frac{2}{Ln(n-1)} \sum_{(i,j)} \Pi_{(i,j)} = \frac{1}{L(n-1)} \left( I - \frac{1}{n} \mathbb{1}\mathbb{1}^\top \right), \quad A^+ = L(n-1)(I - \frac{1}{n}\mathbb{1}\mathbb{1}^\top),$$

where $\mathbb{1}$ is the all-ones vector. Now as $\text{im}(A) = \ker(\mathbb{1})$, this gives

$$\|x\|_A^{*\,2} = L(n-1)\|x\|_2^2 . \tag{14}$$

Similarly, the complete rank-$k$ hypergraph gives $\|x\|_A^{*\,2} = L\frac{n-1}{k-1}\|x\|_2^2$. (Compare to eq. (3.10) and the top of p.21 of [16].) Letting $\mathcal{C}_0 = 4L \max_{x \in X(A):F(x) \leq F(x^0)} \max_{x^* \in F^{\text{arg}}} \|x - x^*\|_2^2$, which is independent of the (hyper)graph as long as it is connected, we thus have a convergence rate of $\frac{n-1}{2}\mathcal{C}_0 \frac{1}{t}$ for the complete graph, and more generally $\frac{n-1}{2(k-1)}\mathcal{C}_0 \frac{1}{t}$ for the complete $k$-graph. Henceforth, we will consider the coefficient in front of $\mathcal{C}_0$ to be the convergence rate.

The above matrix $A$ is a scaled version of what is known as the *graph Laplacian*; given graph $G$ with adjacency matrix $A(G)$ and degree matrix $D(G)$ with the degrees of each vertex on the diagonal, the Laplacian is the matrix

$$\mathcal{L} = \mathcal{L}(G) := D(G) - A(G) . \tag{15}$$

One can check that indeed, $\mathcal{L} = 2 \sum_{(i,j) \in E(G)} \Pi_{(i,j)}$, meaning $A = \frac{p}{2L}\mathcal{L}$, where $p = 1/|E(G)|$ is the uniform probability on edges.

The graph Laplacian is a well-studied object in spectral graph theory and other domains, and we can use existing results to establish bounds for other graphs of interest. To draw this connection, we note two facts: (1) for symmetric matrices $B$, the norm $\langle Bx, x \rangle^{1/2}$ can be bounded by the maximum eigenvalue of $B$, and (2) the maximum eigenvalue of $B^+$ is equal to the inverse of the smallest nonzero eigenvalue of $B$, provided again that $B$ is symmetric.[5] Putting these together, we can therefore bound $\| \cdot \|_A^*$ using the smallest nonzero eigenvalue of $A$, and hence of $\mathcal{L}$. It is easy to see that the smallest eigenvalue is $\lambda_1(G) = 0$ with eigenvector $\mathbb{1}$, and as $G$ is connected, we will have $\lambda_2(G) > 0$. Thus, the smallest nonzero eigenvalue of $A$ is simply $\frac{p}{2L}\lambda_2(G)$, so we have the following for *any* connected graph $G$:

$$\|x\|_A^{*\,2} \leq 2L \frac{|E(G)|}{\lambda_2(G)}\|x\|_2^2 . \tag{16}$$

Of course, by the above definition of $\mathcal{C}_0$ and Theorem 3, this yields the result

$$\mathbb{E}\left[F(x^t) - F^{\min}\right] \leq \frac{|E(G)|}{\lambda_2(G)}\mathcal{C}_0 \frac{1}{t} , \tag{17}$$

showing us how tightly related this eigenvalue is to rate of convergence of Algorithm 1.

The second-smallest eigenvalue $\lambda_2(G)$ is called the *algebraic connectivity* of $G$, and is itself thoroughly studied in spectral and algebraic graph theory. For example, it is known (and easy to check) that $\lambda_2(K_n) = n$, where $K_n$ denotes the complete graph; this together with $|E(K_n)| = n(n-1)/2$ immediately gives eq. (14). In [8], algebraic connectivities are also given for the path on $n$ vertices $P_n$, the cycle $C_n$, the bipartite complete graph $K_{\ell,k}$ for $k < \ell$, and the $k$-dimensional cube $B_k$. We collect these eigenvalues together yields Table 2.

| Graph | $\|V(G)\|$ | $\|E(G)\|$ | $\lambda_2(G)$ |
|---|---|---|---|
| $K_n$ | $n$ | $n(n-1)/2$ | $n$ |
| $P_n$ | $n$ | $n-1$ | $2(1-\cos\frac{\pi}{n})$ |
| $C_n$ | $n$ | $n$ | $2(1-\cos\frac{2\pi}{n})$ |
| $K_{\ell,k}$ | $\ell+k$ | $\ell k$ | $k$ |
| $B_k$ | $2^k$ | $k2^{k-1}$ | $2$ |

Table 2: Algebraic connectivities for common graphs.

Figure 2: Average (in bold) of 30 runs of a separable objective for the complete and star graphs. The empirical gap in iteration complexity is just under 2 (cf. Fig. 3).

Substituting the values in Table 2 into eq. (16), we can directly compare the theoretical convergence rates for different graphs. For example, the star graph $K_{n-1,1}$ has rate $(n-1)(1)/(1) = (n-1)$, which is only a factor of 2 away from the complete graph.[6] The path and cycle fare much worse, yielding roughly $n/2(n^{-2}/2) = n^3$ as $n$ becomes large (applying the Taylor expansion and ignoring $\pi$ terms). Finally, an interesting result due to Mohar [15] says that for any connected graph on $n$ vertices, we have $\lambda_2(G) \geq 4/(n\,\mathrm{diam}(G))$ where $\mathrm{diam}(G)$ is the diameter of $G$. Hence for any connected graph,

$$\mathbb{E}\left[F(x^t) - F^{\min}\right] \leq \frac{n\,|E(G)|\,\mathrm{diam}(G)}{4}\mathcal{C}_0\frac{1}{t}\,, \tag{18}$$

which is useful for sparse graphs of small diameter. See Appendix D.3 for more on hypergraphs.

### D.3 Hypergraphs

Here we briefly show how to analyze general hypergraphs. Representing a hypergraph as a collection $\mathcal{S}$ of hyperedges $S \subseteq [n]$, we may define the degree matrix $D(\mathcal{S})$ to be the diagonal matrix with $D(\mathcal{S})_{ii} = \#\{S \in \mathcal{S} : i \in S\}$, and the "adjacency" matrix to be $A(\mathcal{S})_{ij} = \sum_{S\in\mathcal{S}:i,j\in S} 1/|S|$. Then for uniform probabilities we have $A = \frac{p}{L}(D(\mathcal{S}) - A(\mathcal{S}))$. This follows from observing that for subset $S$, we have $\Pi_S = I_S - \frac{1}{|S|}\mathbb{1}_S\mathbb{1}_S^\top$, and counting as we sum. Taking the complete $k$-graph yields $D(\mathcal{S}) = \binom{n-1}{k-1}I$ and $A(\mathcal{S})_{ij} = \frac{1}{k}\binom{n-2}{k-2} = \frac{k-1}{k(n-1)}\binom{n-1}{k-1}$ for $i \neq j$ and $A(\mathcal{S})_{ii} = \frac{1}{k}\binom{n-1}{k-1}$; putting these together gives $A = \frac{1}{L\binom{n}{k}}\frac{n}{k}\frac{k-1}{n-1}\binom{n-1}{k-1}(I - \frac{1}{n}\mathbb{1}\mathbb{1}^\top)) = \frac{n-1}{L(k-1)}(I - \frac{1}{n}\mathbb{1}\mathbb{1}^\top)$. Similar computations may be done for other hypergraphs of interest.

Figure 3: Thirty runs of a separable objective under the complete and star graphs. The ratio between star and complete of the number of iterations needed to achieve a given objective value is plotted, with the average in bold.

## Footnotes

[3] Note that agents are never risk-seeking; only in the limit as $b \to \infty$ do the traders become risk-neutral.

[4]Everything in this section also holds for a graphical or hypergraphical structure on blocks of coordinates; just add Kronecker products with the appropriate identity matrix.

[5]These facts follow from the operator norm and singular-value decomposition for the pseudoinverse, respectively, together with the fact that singular values are eigenvalues for symmetric matrices.

[6]While of course these are merely upper bounds on the true rates, they match Figures 2 and 3 quite well.