[Reviews · NeurIPS 2015]

Submitted by Assigned_Reviewer_1

Update: The authors provided an interesting response in their final paragraph stating "the matter of whether the behavior model is reasonable is really an economics question." A concise version of this paragraph should be included in the final version in the introduction or at least as a footnote. It helps provide some context as to what problem they are solving.

My biggest questions regard what happens if we compare the predictions of this theoretical analysis to historical pricing data from actual prediction markets? How close does this theory match reality? The results are only in simulated markets.

Clarity issues: 1. The relationship between rho and price should perhaps be given a greater emphasis than be placed in a footnote. The idea of the price of the market is something most readers will be more familiar with than the risk potential functions rho. So, the more explicit the paper the makes the connection to price the more concrete it will be. 2. "This allows us to state informal versions of our main results so as to highlight we address issues of convergence in existing frameworks." should be re-worded. 3. Lines 304-312 are vague. They could use some more explanation. A lot of the math described in words is unclear.

Maybe a couple more citations would help: 1. Line 81 could use a citation 2. '... are "dual" in the same way prices and positions are dual' could use a citation for extra explanation.

Some formatting and notational comments: 1. The authors should study the difference between \citep and \citet and use \citet when appropriate. 2. The notation to use \Delta to refer to the space of distributions should be defined. 3. The inf and sums in (2) and (3) should be explicitly state the variables they operate on. 4. The [N] notation on line 240 is undefined. 5. Some of the proper nouns in the references section are not capitalized. 6. Ref 5 has some formatting issues.

Plotting issues: 1. The y-axis in Figure 1 being termed "Value" isn't very useful. There should a better label. Is it in the units of $? With regard to the x-axis, is # steps = # of trades? 2. Also, maybe the spaghetti plot could be summarized by providing confidence intervals around the averages?
Summary: The authors seek to create a general framework to formalize and understand the convergence of predictions markets. They show an equivalence to a randomized subspace descent algorithm and by showing convergence properties of that optimization algorithm show that predictions markets are efficient in allocating risk.

Submitted by Assigned_Reviewer_2

Summary of paper: The authors consider the rate of convergence to price equilibrium in prediction markets.

They exploit techniques from convex analysis and programming to derive new such bounds for a variety of settings.

In more detail, they consider a potential-based market maker and a number of traders whose preferences are modeled by a risk measure. The market maker can be interpreted as a constant-risk agent.

All players have initial positions in k securities.

Trading dynamics can be naturally interpreted as local rebalancing the positions of a subset of players to minimize their total risk (subject to the sum of their positions remaining the same).

Thus the question of convergence rate is closely related to the question of convergence of gradient descent, where at each iteration only a subset of coordinates are allowed to be updated.

The authors adapt techniques from the latter problems to prove convergence rate bounds for the former problems. Quality: I am in favor of further developing the connections between convex optimization and prediction market design and analysis.

The question of convergence in prediction markets is clearly interesting and, perhaps surprisingly, does not seem very well understood yet.

My one complaint about the paper is that the exact technical statement could be more clearly stated and discussed.

For example, in Theorem 3, it seems that there is a lot of detail hidden within the "constant" R(x^0).

Specifically, this depends not only on x^0, but also on the details of the projections (the \Pi_i's) and how "overlapping" they are.

In the proof of Theorem 3 in the supplementary, the matrix A (used in the norm in (10) and subsequently) is undefined.

Apparently, going by Section D.2, the devil is largely in the details of this A and its properties.

The submission would benefit from a more transparent discussion in the main body about when the convergence rate is good and when it is not.

Clarity: The writing is usually good, though see technical comments above.

Originality: As far as I know, the connection between the convergence analysis of gradient descent variations and prediction markets is original.

Significance: I think this is a significant contribution to the literature on prediction markets.
Summary: The authors consider the rate of convergence to price equilibrium in prediction markets.

They exploit techniques from convex analysis and programming to derive new such bounds for a variety of settings.

Submitted by Assigned_Reviewer_3

This paper presents convergence results in trade network dynamics using convex analysis. In particular, it establishes the equilibrium point of convergence and the rate of convergence, for connected agent networks. The authors model the market dynamics as a modification of block coordinate descent, which they call randomized subspace descent (RSD). Simulation results are provided for complete and star graphs of agent connectivity.

Even though I am not an expert in this field, it appears to me that the results are significant. I also think the paper is fairly well-written (see comments about readability later, though) and is above average for NIPS. I have a few suggestions below.

It is rather odd to call the theorems in Section 2.2 Theorems 4 and 8. I understand that Theorem 4 is made more specific later in Section 4.2, but Theorems 6 & 7 do not even appear in the main paper. I suggest call these lemma or propositions, to avoid this weird numbering.

It seems from the definition of F^{min} that it is a local minimum, rather than the global minimum of the surplus function. If so, this should be highlighted.

Minor comments (to promote readability):

1. There are a plethora of assumptions strewn throughout the paper, some are prerequisites for some analyses, others to further nuance certain claims. It would be better to separately collect these assumptions in a single section upfront. In the current form, it is difficult to evaluate the practicality of the model.

2. Use indices consistently. Sometimes i has been used to index securities, sometimes agents. Best to use distinct indices.

3. Use a distinct notation for vectors, such as boldface or something, to help the reader. Otherwise, oftentimes it is hard to keep track whether something is a vector or a matrix.
Summary: Readability can be improved, but appears to be a fairly strong paper.

Author Feedback
Author rebuttal: We thank all the authors for their suggestions and will fix the typos, undefined or confusing notation, vague graph labels, formatting issues, etc, that were pointed out.

Regarding Reviewer 1's concern about the clarity of the main technical statement, we agree that this can be improved. We will add the definition of the matrix A and unpack the details of the constant R(x^0) just before Theorem 3. Specifically, we will note that R(x^0) increases in: (1) the distance from starting point x^0 to furthest minimizer of F, (2) the Lipschitz constants of F with respect to projections, (3) the connectivity properties of hypergraph induced by the projections.

Reviewer 9's interpretation of the definition of F^{min} suggests that we need to make this clearer in the final version. The range of y in that definition ensures that all possible feasible points are considered and so the minimum is global rather than local. We will add some explanatory notes to this effect.

We agree with Reviewer 9 that the numbering of the Theorems is confusing in its current form. We will find a way to state informal versions of the main theorems so as to make their numbering more straight-forward.

Both Reviewer 9 and Reviewer 5 wonder about the practicality of the model we use and assumptions we make. We would argue that the model and assumptions are practical in the sense that they apply to essentially all potential-based prediction markets in use, for example the LMSR which is used by many platforms. Probably the biggest question then is whether risk-measure-minimizing agents are a realistic behavior model in practice. On the one hand, we are somewhat agnostic to this question, as the two papers posing the open questions which we answer both use (in one, a special case of) this model. That being said, the matter of whether the behavior model is reasonable is really an economics question. Generally speaking, risk-based agents behave like CARA (constant absolute risk aversion) agents in that their risk preferences do not change as a function of total wealth, and thus they are a reasonable approximation to the more broadly accepted CRRA (constant relative risk aversion) model when the wealth levels of the population are bounded. It is an interesting open question whether a more CRRA-like model will still lead to tractable analysis and bounds.